# High Achievers from Low Family Socioeconomic Status Families: Protective Factors for Academically Resilient Students

**DOI:** 10.3390/ijerph192315882

**Published:** 2022-11-29

**Authors:** Yan Yan, Xiaosong Gai

**Affiliations:** School of Psychology, Northeast Normal University, Changchun 130024, China

**Keywords:** socioeconomic status, resilient students, academic resilience, protective factors

## Abstract

Students with low family socioeconomic status (SES) often have lower academic achievement than their peers with high family SES, as has been widely demonstrated. Nevertheless, there is a group of students beating the odds and achieving academic excellence despite the socio-economic background of their families. The students who have the capacity to overcome adversities and achieve successful educational achievements are referred to as academically resilient students. This study’s purpose was to identify the protective factors among academically resilient students. A total of 46,089 students from 303 primary schools in grade 6, 55,477 students from 256 junior high schools in grade 9, and 37,856 students from 66 high schools in grade 11 in a city in northeast China participated in the large-scale investigation. Students completed a structured questionnaire to report their demographic information, psychological characteristics, and three academic tests. A causal comparative research model was applied to determine significant protective factors associated with resilient students (referring to students are resilient if they are among the 25% most socio-economically disadvantaged students in their city but are able to achieve the top 25% or above in all three academic domains). Multivariable logistic regression analyses found that the intrinsic protective factors for resilient students included higher proportion of academic importance identity, higher proportion of achievement approaching motivation, longer-term future educational expectation, and more positive academic emotion compared with non-resilient students; the extrinsic protective factors included parents’ higher proportion of positive expectations for their children’ future development, as well as more harmonious peer and teacher–student relationships. The results of this study provide important targets for psychological intervention of disadvantaged students, and future intervention studies can increase their likelihood of becoming resilient students by improving their recognition of the importance of learning, stronger motivation for achievement approaching, longer-term expectations for future academic careers, and positive academic emotions and harmonious teacher–student relationships.

## 1. Introduction

Students with lower family socioeconomic status (SES) usually have poor academic achievement than their more affluent peers [1,2,3,4,5,6,7]. As family socioeconomic status consists of three parts, financial capital, human capital, and social capital [8], the effect of family socioeconomic status on academic achievement could be divided into three parts. For financial capital, when families have higher incomes, they are able to invest more money in their children’s development which is more likely to be associated with positive academic development [9]. Human capital focuses on the influence of parents’ education resource constraints on education investment and children’s academic development [10]. The study from the perspective of social capital focuses on the influence of social capital inside and outside the family on children’s academic development, represented by parent–child relationship, direct involvement of parents in children’s learning, interaction between parents and schoolteachers and other parents [11]. Research has uncovered a general set of rules in the family environment that can be used to improve students’ academic performance. However, there is very little empirical research on the influencing mechanism of the academic success of disadvantaged students from family socioeconomic status. Some researchers found that the effect of family socioeconomic status is greater than the effect of school factors on academic achievement [3,12]. Poor academic achievement may inhibit the upward social mobility of low-SES students and, thus, may result in the intergenerational transmission of low SES [13]. Modern education systems keep social mobility channels open to all society members from the perspective of equal opportunity in education. Through these transformative characteristics of education, high social status, and a better quality of life channels are open for the socioeconomically disadvantaged individuals [14]. The theory of the positive youth development holds that academic achievement plays an important role in adolescent development [15]. In Chinese culture, academic achievement has always been highly valued and encouraged as a direct way for children to obtain better careers and higher socioeconomic status in the future [16,17]. Therefore, to improve the academic achievement of adolescents with a low family SES, studies have consistently focused on the mitigating factors in the relationship between family socioeconomic status and academic achievement [18,19,20].

However, some adolescents from lower family socioeconomic status are able to overcome their own disadvantaged family environmental factors and achieve higher academic achievement, achieving “academic resilience”. At the most general level, students are academically resilient if they achieve good education outcomes despite their disadvantaged socioeconomic background [21]. Students’ resilience—the odds that a student does well academically despite their disadvantaged background—is operationalized using: (1) the PISA index of economic, social, and cultural status (ESCS) to identify the “adverse circumstances” students experienced, and (2) students’ achievement in the main academic domain in each PISA cycle to identify “good education outcomes” [22,23,24,25]. A substantial number of academic resilience studies rely on the core concern which is “why some students can obtain academic success by breaking through the limitation of their own disadvantages under the same exposure to social and economic adverse conditions, while some students cannot”, and academic resilience reflects the students’ resilience in the field of learning. A student is classified as resilient if she or he is in the bottom quarter of the PISA index of economic, social, and cultural status (ESCS) in the country/economy of assessment and scores in the top quarter of performance among students from all countries/economies, after accounting for socioeconomic status [26].

Kumpfer proposed the framework for resilience in individual–environment interaction [27]. Individuals can achieve resilience by mobilizing their own internal and external factors (protective factors and risk factors) to interact. Internal factors include one’s academic motivation, cognitive ability, social ability, non-cognitive ability, and physical ability; external factors include family, school, peer, and community environmental factors. These factors can interact with each other to reduce the adverse effects of risk factors and thereby increase the likelihood of resilience. Thus, it is clear that the factors that play an important role in academic resilience include both intrinsic the non-cognitive ability and external social factors. Researchers have examined how “resilient students” break out of their disadvantageous situations and achieve higher academic achievement, both at individual and school level. First, at the individual level, students’ education expectations [28,29], self-efficacy and achievement motivation [30], reading metacognitive strategies [31], and supplementary tutoring time [32] were significant predictors of disadvantaged students. Secondly, at the school level, the school’s overall family socioeconomic status level [26,33], school resources [34], more abundant extracurricular activities and school leadership [35], student-centered/inquiry-based instruction [36], and positive school climate [25] were significant predictors of academic resilience outcomes.

Although the aforementioned studies have yielded revealing findings, some studies have relatively small sample sizes, and others have samples from a composite of samples from different regions. Because the type of research design is a causal comparative study, the independent variables are divided by the relationship between family socioeconomic status and academic achievement, and protective factors are used as the dependent variable. Therefore, sample size will directly affect the representativeness of independent variables. Moreover, the family socioeconomic status between different areas or regions lacks direct comparability; the lower socioeconomic status in developed areas may be the same as the middle socioeconomic status in backward areas in absolute value, but psychological feelings may be very different. In order to obtain higher reliability findings, this study investigated the psychological characteristics of resilient students using the data from the large research project on the factors influencing academic quality for all students in grades 6, 9, and 11 in a city of northeast China. A large sample study from the same area ensures that the classification of socioeconomic status and academic achievement grade levels are more precise, and that the determination of the group of resilient students based on the relationship between the two is particularly reliable. The main research question of this study is: What internal and external protective factors are conducive for students from low socioeconomic status families to become high-achieving resilient students? The results of the study will facilitate the identification of psychological intervention targets and help researchers and educators to recognize the need to further effectively guide and assist disadvantaged students and narrow the academic achievement gap between students from high and low socioeconomic status families, thus making education more equitable and fairer for students from all socioeconomic status families and further promoting a balanced development of education in the whole city.

## 2. Materials and Methods

### 2.1. Procedure

Data were collected through educational administration by the structured questionnaires and academic tests in November of 2020. The research protocol was approved by the authors’ affiliated institution, the local education department, and the principal of each school. We obtained informed consent from school administrators, teachers, students, and parents before data collection. Students were told that their participation was completely voluntary and confidential. The study was approved by the authors’ internal approval review board.

### 2.2. Study Samples

Criteria for including and excluding samples were as follow. Samples were included if they met the following criteria: (1) the total score of the Social Desirability Scale was above 2 (seven items, e.g., “Sometimes I gossip about people”; “I never cry”, ranging from 1 = yes, to 2 = no); and (2) the answer rate was above 80%. Ultimately, a total of 46,089 students from 303 primary schools in grade 6, 55,477 students from 256 junior high schools in grade 9, and 37,856 students from 66 high schools in grade 11 were selected from a city in northeast China in December 2020. The selection criteria for resilient students are based on the thematic report issued by OECD in 2018 that designate students with economic, social, and cultural capital status in the lowest quarter of the country and academic achievement in the highest quarter of the country as resilient students. Drawing on the definition of resilient students at international level, resilient students in this study are defined as students whose family socioeconomic status is in the lowest quarter of the city and whose academic achievement is in the highest quarter of the city. The students’ academic achievement was determined by regression analysis after controlling for their family socioeconomic status. There were two exclusion criteria for resilient students: (1) family socioeconomic status was above the lowest quarter of the city; and (2) academic achievement was below the highest quarter of the city. The procedure for identifying resilient students is as follows.

First, the relationship between students’ family socioeconomic status and academic achievement was determined by regression analysis, and students’ family socioeconomic status was used to predict students’ academic achievement.

Second, the difference between the predicted score calculated according to regression equation and the actual score was obtained delta.

In the end, the students were ranked in order, with the highest quartile of delta and the lowest quartile of SES defined as resilient students.

Students were classified by three groups. The first group is those students where the achievement difference was in the highest quarter of the city (residual scores are the difference between the observed score and the expected score) and the lowest quarter of the family socioeconomic status, labeled as resilient students. The second group is those students whose achievement difference was in the lowest quarter of the city and the lowest quarter of the family socioeconomic status, labeled as disadvantaged low achievers. The third group is those students whose achievement difference was in the middle half and the lowest quarter of the family socioeconomic status, labeled as disadvantaged average achievers. The descriptive statistics of the number of people in grade 6, 9, and 11 in the city is shown in Table 1:

As Table 1 shows, in terms of gender ratio in the three grades, the proportion of resilient students was 0.6 percentage points, 14.5 percentage points, and 18.7 percentage points higher among female students than male students, respectively. For family residence, the proportion of resilient students from city and rural areas was slightly higher than that from towns. The proportions of resilient students in grade 6, 9, and 11 in the overal1 valid samples were 7.4%, 9.1%, and 7.7%, respectively; and the proportions of disadvantaged groups were 24.8%, 22.9%, and 22.4%, respectively.

### 2.3. Measures

Family Socioeconomic Status Questionnaire. Family SES was measured based on family economic income, parental education, and parental occupation [37]. Some researchers use parental educational level and parental occupation to measure family socioeconomic status [38]. In this study, we standardized and averaged the parental education and parental occupation to compute an overall measure of family SES. First, parental education was coded according to six categories from 1 (elementary education not completed) to 6 (completed graduate education). Second, parental occupations were assessed according to the classification categories of occupations in China [39]: 1 = jobless, unemployed, and temporary workers, 2 = service and manual 1abor employees, 3 = employees engaged in the transactional work, 4 = employed with no or few employees, 5 = owners of large and medium sized enterprises, 6 = middle managers, 7 = military or police personnel, 8 = professional and technical personnel, 9 = national public officials. The measurement items were converted into z-scores, and then the mean scores were calculated, with higher scores reflecting higher family SES.

Academic achievement. Teachers provided the adolescents’ test scores in three main subjects (i.e., Chinese, math, and English or science). The academic achievement scores were based on objective and time-limited term examinations according to the national curriculum standards for compulsory education students. The original maximum scores for term examinations in Chinese, math, and science were 100 for elementary students, 120 (Chinese, math, and English) for secondary students, and 150 (Chinese, math, and English) for high school students. We standardized Chinese, mathematics, and science or English scores by city and then averaged them to form a single composite academic achievement score. The academic achievement scores ranged from 115 to 85, with higher scores indicating better performance.

Protective Factors. There are 8 items, which can be broadly divided into two categories: intrinsic protective factors (motivation internalization; motivation orientation; educational expectations; academic emotions) and extrinsic protective factors (parental expectation; peer relationships; parent–child relationship; teacher–student relationship). All questions were jointly developed by psychology professors and graduate students based on relevant psychological theories. The eight factors involved in this study are shown in Table 2.

Some researchers have suggested that single-item tests are better if the research aligns with the following criteria: (1) 1arge-sample survey, (2) single-dimensional construct, (3) specific and clear constructs, and (4) time constraints [40,41,42]. As with any psychometric instrument, researchers need to provide convincing evidence from different perspectives to determine the validity of individual measures. First, in terms of consistency of measurement outcomes, in a meta-analysis that synthesized 189 articles on advertising attitudes, it was found that single-item measures of attitudes predicted outcomes almost as well as multi-item measures [43]. Second, for predictive validity, there are studies that support the validity of single-item measures through correlations with outcomes measured either concurrently or subsequently [44,45]. Finally, for criterion validity, the correlation coefficients between the single-item measure of collective efficacy and the mean score of the 20-items measure were found to be *r* = 0.69, *r* = 0.73, and *r* = 0.74 in three sub-studies of a study, respectively [46]. In addition, single-item tests can be used as a viable alternative to increase subjects’ willingness to complete when the researcher wants to focus on as many variables as possible and can minimize subjects’ psychological burden [47]. Therefore, the protective factors of the students in this study were measured using single-item.

### 2.4. Statistical Analyses

Data were analyzed using Microsoft Excel 2010(Microsoft Office, Redmond, WA, USA) and SPSS software version 22.0 (IBM SPSS Statistics, New York, NY, USA). Microsoft Excel was used to edit, sort, and code data; then, the Excel file was imported into SPSS software. Descriptive statistics (frequencies, percentages, means, and standard deviation) and other analyses (i.e., multivariable logistic regression) were performed using SPSS software. Logistic regression was performed with a 95% confidence interval to determine significant associations between the categorical dependent and independent variables. The association of variables was considered statistically significant if the two-sided *p* < 0.05.

## 3. Results

### 3.1. Demographic and Descriptive Statistics

In order to analyze the differences in academic achievement among resilient students, disadvantaged average achievers, and disadvantaged low achievers, one-way analysis ANOVA was conducted with the type of students as the independent variable and the total academic achievement as the dependent variable. The specific results are shown in Table 3.

As Table 3 shows, there was a significant difference (*p* < 0.001) between resilient students, disadvantaged average achievers, and disadvantaged low achievers, and the resilient students had significantly higher academic achievement compared to disadvantaged average achievers and disadvantaged low achievers, whether in grade 6, 9, or 11.

### 3.2. The Comparison of Psychological Characteristics among Resilient Students, Disadvantaged Average Achievers, and Disadvantaged Low Achievers

In order to investigate which factors significantly predict students’ academic resilience in the city, multivariable analyses with predictors entered jointly in the model revealed that most factors remained significant (see Table 4). All variables were included in the analysis as dummy variables. Table 4 shows the multivariable logistic regression analysis results.

As Table 4 shows, first for internal protective factors, the strongest predictor of whether or not to be judged a resilient student was the students’ educational expectation. Students from low socioeconomic status families who expected to complete higher education were 1, 7, and 4 times more likely to be resilient students than those who did not expect to complete higher education in grade 6, 9, and 11, respectively (*p* < 0.001).

Second was academic motivation. For the students in grade 6, 9, and 11, when the intrinsic motivation of disadvantaged students increases by one unit, the likelihood of becoming a citywide resilient student increased by a corresponding 19%, 27%, and 48% (*p* < 0.01, *p* < 0.001, *p* < 0.001); each unit increase in performance-approach goal orientation increased the likelihood of becoming a citywide resilient students by a corresponding increase of 30%, 59%, and 77% (*p* < 0.05, *p* < 0.001, *p* < 0.001).

Finally, it was positive academic emotion. For the students in grade 6, 9, and 11, the likelihood of becoming a resilient student increased by 40%, 17%, and 30% (*p* < 0.001) correspondingly for each unit increase in positive academic emotion among disadvantaged students.

For external protective factors, parents’ expectations for their children’s future development significantly influenced children’s achievement. Parents who expect their children to work for their interests have a more positive impact on their children’s development than parents who care only about their children’s future economic income and social status. Among grade 6 and 9 students, the likelihood of becoming a resilient student increased by 14% and 11% (*p* < 0.05) for each unit increase in positive expectation by parents of disadvantaged students; however, no significant effect of parental expectation of “expecting the children to contribute to society in the future” was found in this study. There was also a protective effect of positive teacher–student relationships each unit increase in positive teacher–student relationships was associated with 20% and 61% increase in the likelihood of being citywide resilient students (*p* < 0.001).

Positive peer relationships also had a protective effect. Among grade 6 and grade 11, each unit increase in the degree of positive peer relationships among disadvantaged students was associated with a corresponding 14% and 21% increase in the likelihood of being citywide resilient students (*p* < 0.05, *p* < 0.001).

## 4. Discussion

Frontier research finds that disadvantaged students tend to have poor academic achievement. However, some groups of disadvantaged students can break through the adverse effects caused by family factors and achieve relatively excellent academic achievement, and they are also known as “resilient students”. This provides us an opportunity to discover what factors can reduce the impact of family socioeconomic status on academic achievement. It also helps researchers and educators to recognize ways and means to further effectively guide and assist disadvantaged students and narrow academic achievement differences between students from with poorer family socioeconomic background and those with better family socioeconomic background [48]. This study first identifies resilient students by two indicators: student background and academic achievement differences, and identify factors that may protect students from various possible initial adverse conditions through regression analysis, especially those that may make schools and teachers take effective action in classrooms and family environments with a higher proportion of students with economic, social, and cultural difficulties and disadvantaged students.

### 4.1. Identifying the Proportion of Resilient Students Is Important

The proportion of resilient students is an important indicator used by international programs to measure educational equity [49]. The focus on resilient students has quantitative implications: first, the proportion of resilient students reflects the current state of equity in educational outcomes in a country (region) and city, as well as providing new perspectives for pedagogical improvement. PISA’s use of the proportion of resilient students to examine equity in educational outcomes is a new perspective that explains the intrinsic link between the development of disadvantaged students and the quality of education and educational equity in a country. Second, this approach extends the study of human capital investment in school contexts at the basic education level. The approach of PISA effectively strips away the influence of other factors, allowing for a more precise exploration of input–output mechanisms.

### 4.2. Factors Affecting Students’ Academic Resilience

First, higher academic expectations increase the likelihood that students will become resilient students. Researchers used data from the Trends in International Mathematics and Science Study (TIMSS) 2011 in Singapore, Korea, Hong Kong, Chinese Taipei, and Japan. The study sample covered 23,354 students in 720 schools in five countries. The study showed that students from disadvantaged families in Singapore who expected to complete two or four years of college were two times more likely to achieve high academic achievement than students from non-disadvantaged families who were expected to be the same (i.e., they were more resilient) [50].The study by Lu Jing [28] obtained similar results to those obtained for resilient students in Singapore, where disadvantaged students who expected to complete their higher education were four times more likely to be resilient students than students who did not expect to complete their higher education sufficiently, in four Chinese provinces and cities. The results of this study also support this idea, with disadvantaged students who expected themselves to complete higher education being 1, 7, and 4 times more likely to be citywide grade 6, 9, and 11 resilient students than students who did not expect themselves to be able to complete higher education. This suggests that the students tested were overly dependent on their own aspirational values rather than environmental factors in the process of academic resilience. This suggests that there is a greater need for career education in China to raise the educational expectations and interest in learning of disadvantaged students.

Secondly, academic motivation is also an important factor in increasing the likelihood of student resilience. In this study, for each unit increase in identity motivation and performance-approach goal orientation among disadvantaged students in 6th, 9th, and 11th grades, the likelihood of becoming a citywide resilient students increased by 19%, 27%, and 48%, and 30%, 59%, and 77%, respectively. Furthermore, researchers have found that achievement motivation has a significant positive effect on disadvantaged students becoming resilient students [51]. Based on self-determination theory [52], their motivation to learn can be divided into external motivation, identity motivation, and external motivation, and the theory emphasizes how much individuals are voluntary or self-determined, it emphasizes the agency of the self in the motivation process, and students’ motivation or student’s agency can have a positive impact on the outcome of learning.

Students’ positive emotions are important in increasing the likelihood of a student being academically resilient. When the positive academic emotion of disadvantaged students increases one unit, the likelihood of becoming citywide resilient students increase by 40%, 17%, and 30%, respectively. Therefore, improving the learning experience of disadvantaged students and creating a relatively positive classroom environment is helpful in fostering students’ interest in learning, motivation, and initiative, which allows students to learn autonomously, happily, and productively, thereby improving learning outcomes.

Researchers analyzed Spanish resilient students in Spain and found that teachers played an important role in helping these students overcome difficulties they may face due to their family socioeconomic background than their peers [53]. This study also found that for each unit increase in the teacher–student relationship, the likelihood of becoming resilient students in the city increased by 20% and 61%. Moreover, for each unit increase in peer relationships among disadvantaged students in grade 6 and 11, the likelihood of becoming resilient students in the city increased by 14% and 21%.

Finally, among grade 6 and 9 students, the likelihood of becoming a citywide resilient student increased by 14% and 11% for each unit increase in parental interest-based expectations of disadvantaged students; high parental expectations, increased educational expenditures, and positive perceptions of self appeared to be strong predictors of adolescent cognitive ability, which is consistent with previous findings. Parents’ expectations for children’ education is particularly salient in Asian cultures [54]. Researchers have found that high parental expectations of education are associated with more positive learning behaviors among students in Asian families [55]. Similarly, a study by Mau [56] reported that Asian immigrants and Asian American tenth graders who perceived higher educational expectations from their parents were likely to invest more effort in academic achievement. In addition, some studies have found that only parents and children have consistent high educational expectations, can make children get expectations and care, strengthen the motivation to achieve expectations while being encouraged, and then promote students’ learning behavior to a certain extent, and ultimately have a positive impact on academic achievement [57]. This shows that parents’ educational expectations and students’ educational expectations are very important, but it is important to pay attention to the consistency between them.

The results of the study found that resilient students showed higher percentages of identity motivation, higher educational expectations, more interest in doing what they like, and higher percentages of positive academic emotions compared to disadvantaged average achievers and disadvantaged low achievers. The degree to which students’ academic achievement is related to family socioeconomic status reflects educational balance, meaning that the lower the degree to which students’ academic achievement is influenced by family socioeconomic status, the higher the degree of educational balance [58]. These findings provide important insights for reducing the influence of family socioeconomic status on academic achievement, developing students’ resilience targets, and promoting balanced development of education.

### 4.3. Educational Recommendations

It is found that academic expectation and learning motivation are very important to improve the possibility of disadvantaged students becoming resilient students, whether in the grade 6, 9, or 11. Therefore, it is necessary to combine scientific methods to stimulate students’ learning motivation and establish positive academic expectation. The teachers and psychology teachers in schools can incorporate career education and develop growth mindset development programs to stimulate students’ own intrinsic motivation, enhance their learning efficacy and motivation, and thus improve their academic achievement.

Parents could fully understand the importance of their children’s educational expectations, create flexible communication with their children, and promote the formation of positive feedback.

Teachers and psychology teachers in schools can cultivate students to maintain positive academic emotion, so that students will be more active in teaching activities. The positive academic emotion not only helps to shape students’ positive and optimistic personality attitude, but also helps to enhance students’ enthusiasm for academic activities, and finally achieve the improvement of students’ academic achievement.

Schools could actively improve students’ learning conditions, especially to create convenience for students’ access to knowledge and information, constantly open students’ horizons, promote students’ self-thinking, and help students form internal motivation for self-improvement.

### 4.4. Limitations and Perspectives

First, while this study contained many personal variables, only a few family- and school-level factors were explored. School factors are very complex, and it has been found that school factors such as high quality school learning activities [58], school quality [59], and school average family socioeconomic status [60] may be one of the ways to compensate for the adverse effects caused by the lack of family education resources in disadvantaged students. Therefore, future studies should add more investigation of school variables in the study in order to better understand the root cause of academic resilience, and more school samples should be selected from future studies for multilevel model analysis. This study included only variables from family, teacher–student relationship, peer relationship, and some individuals. If there are more variables at these levels, it will be more meaningful to explain and help researchers better understand the more critical factors for students’ academic success.

Second, this study only uses cross-sectional data to explore the approach of academic resilience in adolescents; therefore, causal inferences could not be drawn. Further longitudinal investigations and more in-depth qualitative studies will be highly warranted.

Third, considering intellectual ability may influence academic achievement, and intellectual ability is one of important predictors for academic resilient students; therefore, it is important for future research to investigate the influence of the intellectual ability on the academic resilient students.

## 5. Conclusions

Students who are motivated by the ‘learning is important’ identity, motivated by performance-approach goal orientation, have higher educational expectations, have positive peer relationships, have positive teacher–student relationships, and hold interest-based parental expectations are more 1ikely to be resilient students.

## Figures and Tables

**Table 1 ijerph-19-15882-t001:** Descriptive statistics of disadvantaged students in each grade.

Grade	Variable		Resilient Students		Disadvantaged Average Achievers		Disadvantaged Low Achievers	
			*n*	%	*n*	%	*n*	%
6th	Gender	Male	1688	49.5	3197	46.5	1653	47.6
		Female	1706	50.1	3636	53	1797	51.8
		Missing	12	0.4	31	0.5	21	0.6
	Residence	City	1236	36.3	2238	32.6	912	26.3
		Town	650	19.1	1337	19.5	534	15.4
		Rural	1498	44	3238	47.2	1975	56.9
		Missing	22	0.6	51	0.7	50	1.4
9th	Gender	Male	2144	42.6	5271	46.1	3069	56.1
		Female	2867	57.1	6080	53.3	2342	42.8
		Missing	14	0.3	66	0.6	55	1
	Residence	City	1508	30	2804	24.6	1300	23.8
		Town	1028	20.5	2208	19.3	892	16.3
		Rural	2471	49.1	6348	55.6	3210	58.7
		Missing	18	0.4	57	0.5	64	1.2
11th	Gender	Male	1180	40.6	2680	38.7	1770	56.3
		Female	1721	59.3	4226	61.1	1366	43.4
		Missing	2	0.1	13	0.2	11	0.3
	Residence	City	964	33.3	1747	25.3	696	22.2
		Town	701	24.1	1622	23.4	671	21.3
		Rural	1229	42.3	3519	50.9	1760	55.9
		Missing	9	0.3	31	0.4	20	0.6

**Table 2 ijerph-19-15882-t002:** Description of variables.

Variable Categories	Variable	Description
Intrinsic protective factors	Motivation Internalization	What’s the reason for your hard work? Identify motivation (Learning is important); Intrinsic motivation (Learning is fun); Extrinsic motivation (The rewards and punishments of others)
	Motivation Orientation	What’s the reason for you listen to the teacher carefully at school? Performance-approach goal orientation (Get a good ranking or praise); Mastery approach goal orientation (Increase knowledge and ability); Performance-avoidance goal orientation (Afraid of getting criticized for not doing well)
	Educational Expectations	What grade do you expect yourself to complete? Middle school/high school graduate. High school/college graduate, College/University graduate. Postgraduate
	Academic Emotions	What emotion do you most often experience during study activities? Positive emotions (Happy, Proud, Fun); Negative emotions (Stressed, tired, bored, sad)
Extrinsic protective factors	Parental Expectations	What kind of person your parents want you to be in the future? Interest-based expectation (To do what interests them); Contribution based expectation (Become a contributor); Utilitarian expectation (Be an official or make a lot money)
	Peer Relationship	How do you get along with your classmates in the class? Positive acceptance type (My classmates like to be with me); Rejects type (Some of my classmates refused to let me join them); Controversial type (Some people like to play with me, some people don’t like); Ignoring type (My classmates don’t pay much attention to me)
	Parent–Child Relationship	Which of the following statements best describes your relationship with your parents? Healthy type (Parents listen to their children carefully and respect their children’s decisions); Dependent type (Parents make decisions on all the things); Alienative type (Children make their own decisions, regardless of their parents’ feelings and opinions)
	Teacher–Student Relationship	What is your relationship with your teacher? Intimate type (I like my teachers); Rejects type (The teachers don’t like me); Controversial type (Often have trouble with teachers); Ignoring type (The teacher didn’t pay much attention to me)

**Table 3 ijerph-19-15882-t003:** The comparison of academic achievement among resilient students, disadvantaged average achievers and disadvantaged low achievers.

	Resilient Students		Disadvantaged Average Achievers		Disadvantaged Low Achievers		*F*	$\eta_{p}^{2}\;$
	*M*	*SD*	*M*	*SD*	*M*	*SD*		
6th	116.03	5.41	97.47	5.97	78.71	6.39	33,834.44 (*p* < 0.001)	0.83
9th	115.21	5.08	95.85	6.55	79.04	4.39	51,698.45 (*p* < 0.001)	0.83
11th	116.11	5.74	96.16	6.67	79.82	3.51	29,194.37 (*p* < 0.001)	0.82

**Table 4 ijerph-19-15882-t004:** Multivariable logistic regression analysis of students’ resilience.

Variable	Grade 6		Grade 9		Grade 11	
	*OR* (95%*CI*)	*p*-Value	*OR* (95%*CI*)	*p*-Value	*OR* (95%*CI*)	*p*-Value
Gender						
Male	1.22 (1.12–1.32)	<0.001	1.05 (0.97–1.13)	0.217	1.12 (1.01–1.23)	<0.05
Female						
Motivation Internalization						
Identify motivation	1.19 (1.04–1.36)	<0.01	1.27 (1.14–1.41)	<0.001	1.48 (1.3–1.69)	<0.001
Intrinsic motivation	1.04 (0.91–1.2)	0.576	0.92 (0.82–1.05)	0.218	1.18 (0.99–1.42)	0.072
Extrinsic motivation						
Motivation Orientation						
Performance-approach goal orientation	1.30 (1.07–1.58)	<0.05	1.59 (1.35–1.88)	<0.001	1.77 (1.41–2.22)	<0.001
Mastery approach goal orientation	1.16 (0.97–1.39)	0.097	1.1 (0.94–1.3)	0.240	1.07 (0.85–1.34)	0.585
Performance-avoidance goal orientation						
Educational Expectations						
High school/college graduate	1.03 (0.72–1.48)	0.855	0.73 (0.52–1.02)	0.063	0.37 (0.22–0.61)	<0.001
College/University graduate	1.61 (1.18–2.21)	<0.05	2.43 (1.8–3.27)	<0.001	1.15 (0.8–1.66)	0.454
Postgraduate	2.20 (1.61–3.02)	<0.001	8.24 (6.11–11.1)	<0.001	5.06 (3.53–7.27)	<0.001
Middle school/high school graduate						
Academic Emotions						
Positive emotions	1.40 (1.28–1.54)	<0.001	1.17 (1.09–1.27)	<0.001	1.30 (1.18–1.44)	<0.001
Negative emotions						
Parental expectations						
Interest based expectation	1.14 (1.01–1.28)	<0.05	1.11 (1.00–1.22)	<0.05	1.09 (0.96–1.23)	0.174
Contribution based expectation	1.07 (0.94–1.21)	0.296	1.07 (0.96–1.21)	0.226	1.17 (1.00–1.37)	0.05
Utilitarian expectation						
Peer Relationship						
Positive acceptance type	1.14 (1.05–1.24)	<0.05	1.03 (0.96–1.11)	0.43	1.21 (1.10–1.33)	<0.001
Other type						
Parent–Child Relationship						
Healthy type	0.97 (0.88–1.06)	0.498	1.08 (0.99–1.17)	0.074	1.07 (0.96–1.19)	0.213
Other type						
Teacher–Student Relationship						
Intimate type	1.20 (1.09–1.33)	<0.001	1.61 (1.46–1.77)	<0.001	0.91 (0.72–1.16)	0.465
Other type						

## Data Availability

Further inquiries about the data supporting reported results can be directed to the corresponding author.

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
