# Peer review of "High Achievers from Low Family Socioeconomic Status Families: Protective Factors for Academically Resilient Students"

_ijerph, 2022, doi:10.3390/ijerph192315882_

Round 1

Reviewer 1 Report

This is a meaningful study that explored the protective factors for academically resilient students from different families with different social-economic status. This study is well-designed and developed. However, I still have the following suggestions.

Introduction:

I think the introduction is too short and needs to be developed further. I can add some background introduction and details of the research gap.

Materials and Methods

When did the survey conduct?

I am not sure whether the authors considered the impact of COVID-19 pandemic on the data collection and analysis of the data.

Please mention the inclusion and exclusion criteria of the study samples in detail.

Discussion:

The results should be further discussed with reference to the prior studies reviewed in the literature review part. For example, authors can give more explanations for using protective factors to protect low-family socioeconomic status students

Last but not least, there are a lot of typos within the manuscript. I strongly hope the authors can ask a native speaker to polish this paper before resubmitting the revised version.

Author Response

We sincerely appreciate the time and effort that you have spent in reviewing our manuscript, and we thank you for your valuable suggestions and comments. Our revisions are indicated in red font in the revised manuscript. We hope that our revisions will be met with approval. If further clarification is needed, we would be happy to make additional modifications. Thank you very much for your careful reading again. 

Response1:We have revised the Introduction already. (See page 1-2 in red)

Response2: Thank you for your careful review. Our Data were collected through  questionnaires in December 2020. Since December 2020,the data of zero COVID-19 cases in Jilin province for 214 days. So we think the COVID-19 pandemic did not impact.  The inclusion and exclusion criteria of the study samples have revised already. (See page 3 in red)

Response3: We have revised the Educational Recommendations already. (See page 11 in red)

Other details please see the attachment.

Reviewer 2 Report

Thank you very much for giving me the opportunity to review the manuscript entitled “High achievers from low family socioeconomic status families: protective factors for academically resilient students”. I have reviewed it with great interest. While I agree that the topic of this study is very important and interesting, I have some concerns with the manuscript in its current form. First, I feel that intellectual ability is one of important predictors. Intrinsic protective factors (motivation internalization, motivation orientation, educational expectations, and academic emotions) and extrinsic protective factors (parental expectation, peer relationships, parent-child relationship, and teacher-student relationship) were examined as predictors of students’ academic resistance in the city. It is clear that students with lower levels of intellectual ability are less likely to be academically resilient. I appreciate if you would explain why intellectual ability has not been considered as a predictor. Second, the terms “resilience” and “resistance” are used interchangeably. It seems better to unify these terms.

Also, I have list up minor points.

1) Table 1. It seems better to put space between lines so that readers can understand the results.

2) Table 3. Some words and numbers are bold.

3) Page 3, 2.2. Study Samples. It seems better to describe the inclusion and exclusion criterion.

4) Page 6, 2.4. Statistical Analyses. It seems better to clarify reference for each predictor.

5) Line 382, “Second, this study only uses cross-sectional data to explore the approach of academic resistance in adolescents”. It seems better to describe limitations of the cross-sectional design more specifically.

Author Response

We sincerely appreciate the time and effort that you have spent in reviewing our manuscript, and we thank you for your valuable suggestions and comments. Your insight suggestions and comments improve the reliability of our manuscript.  First, for Intelligence, as this study using tens of thousands of people sample data, lack of labor resources for large-scale intelligence tests. Second, your opinion is very enlightening. We are agree with your opinion that  students with lower levels of intellectual ability are less likely to be academically resilient. This is one of the limitations of this study. We have revised this part in the limitations, and we will put your insight suggestions for future research.  Third,  we have revised the terms “resistance” into “resilience” in red font. Our revisions are indicated in Red font in the revised manuscript.  We hope that our revisions will be met with approval. If further clarification is needed, we would be happy to make additional modifications. Thank you very much for your careful reading again. Other details please see the attachment.

Round 2

Reviewer 2 Report

Thank you very much for your responses to my comments. I have confirmed that my comments had been addressed.